# An Emerging Role for Calcium Signaling in Cancer-Associated Fibroblasts

**DOI:** 10.3390/ijms222111366

**Published:** 2021-10-21

**Authors:** Francisco Sadras, Gregory R. Monteith, Sarah J. Roberts-Thomson

**Affiliations:** 1School of Pharmacy, The University of Queensland, Brisbane, QLD 4072, Australia; f.sadras@uq.edu.au (F.S.); greg@pharmacy.uq.edu.au (G.R.M.); 2Mater Research, Translational Research Institute, The University of Queensland, Brisbane, QLD 4072, Australia

**Keywords:** calcium, cancer, cancer-associated fibroblasts, migration, chemoresistance

## Abstract

Tumors exist in a complex milieu where interaction with their associated microenvironment significantly contributes to disease progression. Cancer-associated fibroblasts (CAFs) are the primary component of the tumor microenvironment and participate in complex bidirectional communication with tumor cells. CAFs support the development of various hallmarks of cancer through diverse processes, including direct cell–cell contact, paracrine signaling, and remodeling and deposition of the extracellular matrix. Calcium signaling is a key second messenger in intra- and inter-cellular signaling pathways that contributes to cancer progression; however, the links between calcium signaling and CAFs are less well-explored. In this review, we put into context the role of calcium signaling in interactions between cancer cells and CAFs, with a focus on migration, proliferation, chemoresistance, and genetic instability.

## 1. Introduction

### 1.1. Cancer and the Tumor Microenvironment

Cancer is a highly heterogeneous disease and is one of the primary causes of death worldwide. Many studies and comprehensive reviews have linked calcium (Ca^2+^) signaling with cancer progression [1] and detailed the role of Ca^2+^ in intercellular communication, primarily in the nervous system [2,3] but also in cancer [4,5]. The interplay between cancer cells and their surrounding environment, termed the tumor microenvironment (TME), and especially cancer associated fibroblasts (CAFs), has been of growing interest over the past decade. Despite published evidence for the role of Ca^2+^ signaling in this crucial TME compartment [6,7,8], there are no reviews on Ca^2+^ signaling in CAFs. This review discusses the association between CAFs and cancer cells in the context of Ca^2+^ signaling, as well as Ca^2+^ signaling in CAFs, and how these impact both cancer progression and the development of CAF properties.

### 1.2. Cancer-Associated Fibroblasts

There has been a growing focus on understanding the communication and interplay between cancer cells and the TME and how these interactions contribute to tumor progression [9,10]. The TME consists of diverse cell types, including fibroblasts, adipocytes, immune cells, and endothelial cells, as well as the extracellular matrix (ECM) [9]. Together, the TME significantly affects most, if not all, of the hallmarks of cancer [11]; hence, understanding the role of the TME in tumor progression has clear prognostic and therapeutic implications.

Of the TME components, the most studied, and often the most abundant, are CAFs [12]. CAFs can arise from a variety of cell types, including fibroblasts, adipocytes, and endothelial cells that, through signals from the tumor, are transformed from a tumor-suppressive to a tumor-promoting CAF phenotype [13]. As CAFs are derived from distinct cellular populations, there is no single CAF marker; however, there are common markers, such as α-smooth muscle actin (αSMA), fibroblast activation protein (FAP), fibroblast-specific protein 1 (FSP1), and the platelet-derived growth factor receptor (PDGFR), that are often expressed by CAFs but not by their normal counterparts [14].

CAFs contribute to numerous cancer hallmarks through a variety of mechanisms, including paracrine factor secretion, modifying the ECM, and direct physical contact [14]. As well as the broad contributions of CAFs towards cancer progression, chemoresistance [15], proliferation [16], migration [17], CAF differentiation [18], and genetic instability [19] are linked with Ca^2+^ signaling in CAFs and cancer. To date, the literature has focused on the role of Ca^2+^ signaling in migration for CAF–cancer cell interactions. However, given the extensive links of Ca^2+^ signaling in diverse cancer phenotypes, we suggest it is likely that additional roles for Ca^2+^ signaling in CAF–cancer interactions will be revealed as the field matures.

### 1.3. Ca^2+^ Signal

Ca^2+^ is a ubiquitous second messenger that controls diverse cellular functions in both physiological and pathological settings [1,20]. Ca^2+^ signaling can be initiated by endogenous and exogenous mechanisms and, consequently, is frequently involved in intercellular communication either through direct transfer of Ca^2+^ through gap junctions and nanotubes [21,22,23] or by signal transduction pathways, such as G protein-coupled receptors (GPCRs) or tyrosine kinase receptors (TKRs) [3,24]. The Ca^2+^ gradient in cells is tightly controlled, with dramatically different concentrations across subcellular compartments and the extra-cellular fluid. Resting cytoplasmic Ca^2+^ concentration ([Ca^2+^]_CYT_) is ~100 nM, while the extra-cellular fluid is >1 mM [1], with the primary intracellular store of the endoplasmic reticulum falling between these at 100–800 µM [25,26]. As a result of these significant concentration gradients, Ca^2+^ can enter the cytoplasm through open channels and facilitated diffusion, while it requires energy to be actively pumped from the cytoplasm into other cellular organelles by Ca^2+^ ATPases, such as those of the Sarcoplasmic/Endoplasmic Reticulum or out of the cell by plasma membrane Ca^2+^ ATPases [1]. In addition to calcium entry into the cytoplasm, intra-organelle communication is an emerging field and is one area that should be the focus of future studies. To date, studies have primarily been conducted on cancer cells [27], while fewer have been done on CAFs.

Three key pathways for cytoplasmic Ca^2+^ entry include voltage-gated calcium channels (VGCC), transient receptor potential calcium channels (TRP), and store-operated calcium entry (SOCE) [2]. VGCCs are activated by depolarized membrane potentials [28,29], while TRP channels have a wide variety of activators ranging from temperature to mechanical stimuli to endogenous and exogenous chemicals [30,31,32]. Compared to these channels, SOCE is activated by a two-step process: first the internal endoplasmic reticulum (ER) stores are released, typically downstream of GPCR or TKR activation, and then this depletion of ER calcium is detected by stromal interaction molecule 1 (STIM1), which activates the ORAI1 channel, resulting in Ca^2+^ entry [1]. The ER stores are then refilled by the Sarcoplasmic/Endoplasmic Reticulum Calcium ATPase (SERCA). Of these three pathways, SOCE is the predominant pathway of calcium signaling in non-excitatory cells.

Depending on the strength and temporal and spatial resolution of the signal, increases in [Ca^2+^]_CYT_ can result in a myriad of outcomes, including gene transcription and protein degradation, proliferation, and cell death [1]. These crucial and varied roles for Ca^2+^ in cellular physiology have been thoroughly demonstrated in cancer, with Ca^2+^ signaling being critical in cancer proliferation [33], apoptosis [34,35], and chemoresistance [24,32]. Despite these extensive findings on Ca^2+^ signaling in cancer, studies on the role of Ca^2+^ signaling in that most crucial of stromal components, CAFs, are few and fragmented. Here, we consolidate current research on the bidirectional interactions between CAFs and cancer cells through the lens of Ca^2+^ signaling in the context of phenotypic changes in cancer cells and CAFs.

## 2. CAF–Tumor Ca^2+^ Signaling

CAFs are well-established as modulators of tumor phenotype through a variety of mechanisms [9]. This section presents a comprehensive summary of the current literature on Ca^2+^-mediated CAF–cancer communication. We cover two broad categories: chemoresistance and proliferation, migration, invasion, and metastasis (Figure 1).

### 2.1. Tumor Chemoresistance and the Importance of Stroma

Apoptosis and chemotherapeutic screens are often initially performed in vitro and can provide valuable information on novel therapeutic targets. However, a significant number of lead compounds fail upon in vivo studies. This mismatch between in vitro and in vivo assays is partially associated with over-simplified in vitro systems that assess cancer cells in monoculture. An intermediate step between monoculture in vitro studies and more relevant in vivo assays is to use a co-culture system that incorporates some of the complexity present in vivo.

One example of a potential therapeutic that has differing roles in vivo and in vitro is resveratrol. In vitro resveratrol inhibits tumorigenesis in various cancer types, including breast, prostate, and colon cancer [36,37,38]. When resveratrol is administered in vivo, however, results are ambiguous, and responses vary based on route of administration, dose, tumor model, species, and other factors that can be difficult to identify [39]. An in vitro study that focused on the interactions between prostate cancer cells and CAFs identified that resveratrol activates the TRPA1 calcium channel in CAFs and leads to a strong Ca^2+^ influx and secretion of VEGF and HGF [8] (Figure 1A). TRPA1 activation significantly reduces prostate cancer cell death by 40% in a co-culture model. Use of HC-030031, a TRPA1 inhibitor, significantly reduced the Ca^2+^ influx and VEGF and HGF secretion in CAFs and blocked the protective effect on prostate cancer [8]. This study highlights the value of using co-culture alongside monoculture models for assessing chemotherapeutics and potential roles for Ca^2+^. Ongoing work is improving simple co-culture models to more accurately reflect the complexity observed in vivo by using microfluidics and patient-derived samples [40], as well as in vivo models examining the roles of CAFs on chemoresistance through interactions with other components of the TME [41].

### 2.2. Tumor Proliferation, Migration, Invasion, and Metastasis

Calcium signaling is an established driver of tumor proliferation, migration, invasion, and metastasis [1]. Studies examining CAF–cancer interactions have revealed that CAFs effect changes on these same properties in a Ca^2+^-dependent manner, as described below.

Environmental factors are well-known to contribute to cancer progression [41]. Triclosan (TCS) is an antimicrobial agent widely used in personal care products that can deregulate hormone production [4]. Like resveratrol, TCS can induce TRPA1-mediated Ca^2+^ influx in prostate cancer-derived CAFs, leading to increased secretion of VEGF (Figure 1A). VEGF secretion and Ca^2+^ influx were blocked by the TRPA1 inhibitors HC-030031, AP-18, and A967079 [4]. Treatment of prostate adenocarcinoma PC3 cells with TCS-induced CAF-conditioned media promoted PC3 proliferation, and this phenotype was also blocked by TRPA1 inhibitors. These studies provide further support for a general role for TRPA1-mediated VEGF secretion in CAFs promoting tumor growth in a Ca^2+^-dependent manner.

C-X-C chemokine receptor type 4 (CXCR4) is a well-studied chemokine receptor that contributes to migration and invasion in diverse settings in a Ca^2+^-dependent manner [42,43]. CXCR4 is canonically activated by Stromal Cell Derived Factor 1 (SDF-1), including CAF-secreted SDF-1 [44,45,46]. CXCR4 activation in cancer cells promotes Ca^2+^-dependent migration [44,45,46], possibly through PI3K/Akt and MAPK/Erk signaling [44]. Three studies have linked CAF-secreted SDF-1 with activation of CXCR4 and increased migration and invasion in endometrial and breast cancer (Figure 1B) [44,45,46]. While these papers do not directly show a role for Ca^2+^ in CXCR4 signaling, Chao et al. [47] found that CAFs had elevated expression of SDF-1 and that exogenous SDF-1 activates a Ca^2+^ response and increases anchorage-independent growth in CD133+/CXCR4+ colon cancer cells. However, this study did not show that CAF-secreted SDF-1 could produce this outcome and used exogenous addition of SDF-1. Combined, these findings show that (A) CAFs produce and secrete SDF-1, which promotes migration and invasion in cancer cells, and (B) SDF-1 produces a Ca^2+^ response that is required for the migration phenotype (Figure 1B).

CAF-secreted matrix metalloproteases (MMP) are well-established as extra-cellular matrix re-modelers and correlate significantly with poor prognosis [6,48,49]. Protease-activated receptors (PARs) are tethered-ligand receptors that, upon proteolytic cleavage of their extracellular domains, release their tethered ligand into the cytoplasm, which triggers multiple signaling pathways, including Ca^2+^ influx. CAF-secreted MMP-1 cleaves PAR1 present in breast cancer cells and generates Ca^2+^ signals that promote migration (Figure 1C) [50]. Inhibition of either MMP-1 or PAR1 significantly decreases tumor growth, implicating CAF-secreted MMP-1 and PAR1-mediated Ca^2+^ influx in breast cancer progression.

CAFs are routinely generated either through surgical extraction from cancer patients [51] or by activation of normal fibroblasts using agents such as TGFβ1 [52] or complement [53]. Liu et al. [54] used a different approach and converted mammary reduction fibroblasts into senescent fibroblasts using hydrogen peroxide. With the rationale that cancer incidence increases with age and is typically associated with increased genetic instability in precancerous cells, there may also be changes in TME cells that have undergone senescence, in turn promoting tumorigenesis. Senescent fibroblasts degrade T-Lymphoma Invasion and Metastasis-Inducing Protein 1 (TIAM1) in a Ca^2+^-dependent manner (Figure 1D) [54]. TIAM1 degradation increases levels of secreted osteopontin (OPN), which increases migration and invasion in human mammary epithelial cells (Figure 1D) [47]. Senescent fibroblasts can thus facilitate the onset of cancer hallmarks in normal mammary cells in a Ca^2+^-dependent manner.

## 3. Tumor–CAF Ca^2+^ Signaling

CAFs modulate cancer cell properties, and, conversely, tumor cells modify their microenvironment as a critical part of tumor progression, leading to the development of CAFs [9]. Here we describe the effects of tumor-mediated Ca^2+^ signaling in CAFs in four areas: differentiation; proliferation, migration, and invasion; genetic instability; and wound healing (Figure 2).

### 3.1. CAF Differentiation

CAFs are routinely derived from normal fibroblasts using a variety of activators, principally TGFβ1 [13,52]. To uncover a potential role for Ca^2+^ signaling in TGFβ1 mediated differentiation, we examined functional Ca^2+^ signaling and calcium channel expression levels in CAFs [18]. Store-operated calcium entry (SOCE) is a major mechanism for refilling internal Ca^2+^ stores and can induce a wide range of transcriptional and phenotypic changes. SOCE-mediated Ca^2+^ flux was significantly reduced following induction of a CAF phenotype in breast fibroblast cells [18]. Additionally, two voltage-gated calcium channels, Ca_V_1.2 and Ca_V_3.2, were upregulated in both our in vitro CAF model and in a group of patient-derived breast CAFs. Pharmacological or siRNA inhibition of these channels significantly impaired CAF activation in vitro (Figure 2A) [18]. While these studies did not use cancer cell-secreted TGFβ, this is a commonly secreted signaling molecule in breast cancer [55], indicating this transformation is likely to occur in vivo.

### 3.2. CAF Proliferation, Migration, and Invasion

Similar to cancer cells displaying aberrant proliferation, migration, and invasion phenotypes compared to their normal counterparts, these properties change during fibroblast differentiation into CAFs. Here we present examples on how cancer cells can induce these changes in CAFs through Ca^2+^ signaling.

Pancreatic stellate cells (PSCs) are the most common CAF subtype in pancreatic ductal adenocarcinomas (PDAC), and their presence correlates with poor survival [56]. Like other CAF types, PSCs often display increased migration after exposure to PDACs [57] and can co-migrate with PDAC cells, similar to the behavior of squamous cell carcinoma [58]. Mechanistically, this increased migration in PSCs after culture with PDAC-conditioned media is through the activation and cooperation of a potassium channel (K_Ca_3.1) and a calcium channel (TRPC3). PDAC-conditioned media significantly increase migration and Ca^2+^ influx in PSCs, and either pharmacological inhibition of K_Ca_3.1 or siRNA knockdown of TRPC3 blocked this migration, indicating that both TRPC3 and K_Ca_3.1 channels and Ca^2+^ influx were required for PDAC-mediated migration in PSCs (Figure 2B) [57].

The “endothelial retraction factor”, 12(S)-HETE [59], facilitates neutrophil and monocyte transmigration through endothelial cell barriers by downregulating VE-cadherin expression. In cancer, it has a similar role and facilitates the migration of breast and colon cancer cells through endothelial barriers, promoting metastasis [60,61]. CAFs present a different type of barrier that cancer cells must overcome to successfully disseminate. Interestingly, a similar phenomenon to this endothelial retraction was demonstrated with colorectal cancer cells and CAFs. Colorectal cancer cells secrete 12(S)-HETE, which promotes CAF retraction from colorectal cancer spheroids by activating myosin light chain 2 (MLC2) through Ca^2+^/RHO/ROCK signaling axis, facilitating colorectal cancer cell migration into the stroma and surrounding regions (Figure 2C) [7]. This demonstrates a novel 12(S)-HETE signaling pathway that promotes retraction through MLC2 without affecting VE-cadherin. Two additional studies demonstrated a role for MLC2 in CAF-mediated ECM remodeling and contractility through RHO/ROCK signaling [62,63] but did not examine the potential role of calcium.

Estrogen is commonly secreted in physiological and pathological breast tissue, and the presence of the estrogen receptor is critical for prognosis and treatment [64]. While most studies have focused on the canonical receptor and its expression in cancer cells, recent research has shown that breast CAFs express the G-protein-coupled estrogen receptor (GPER) [65,66]. CAFs expressing GPER exposed to estrogen or the GPER agonist G1 display increased intracellular Ca^2+^ and increased proliferation and migration through GPER/EGFR/ERK signaling. This provides circumstantial evidence for a link between GPER-mediated Ca^2+^ signaling and proliferation and migration in CAFs (Figure 2D). While these studies did not directly assess the effect of increased intracellular Ca^2+^ on CAF migration and proliferation, the requirement of Ca^2+^ signaling for these processes is well-documented [20].

### 3.3. CAF Genetic Instability

Microbeam irradiation technique is a major advance that allows a selected fraction of cells within a population to be targeted at specific subcellular locations with a precise dose of radiation [67]. Targeted irradiation is particularly helpful for cancers affecting the CNS, as it can limit the damage to surrounding cells. A potential complication from irradiation therapy is genetic instability arising from the bystander effect. Genetic instability can contribute to metastasis and chemoresistance by increasing genetic diversity within a tumor population [68]. Microbeam irradiation has been used to study the bystander effect, where cells that are near irradiated cells may undergo phenotypic changes [69]. The bystander effect in glioma cells and their associated fibroblasts was studied by selectively irradiating glioma or fibroblast cells in a co-culture model or using a conditioned medium from irradiated cells [70]. In all scenarios, a Ca^2+^ influx was triggered in both fibroblasts and glioma that was associated with the formation of micronuclei (Figure 2E). Micronuclei formation was dependent on Ca^2+^ signaling through VGCCs as treatment with calcicludine, an L-type VGCC blocker, inhibited the formation of micronuclei. Increased genetic instability in neighboring CAFs and glioma post microbeam irradiation highlights the importance of selective targeting.

Apoptosis is common in tumors and can be induced by internal and external stimuli [71]. The resulting apoptotic bodies can be cleared by both tumor cells engulfing the apoptotic bodies and by stromal cells through a process termed efferocytosis [72]. Efferocytosis is enabled by both professional phagocytes, such as macrophages and mast cells, and non-professional phagocytes, including fibroblasts [72]. Human papillomavirus (HPV) is primarily associated with the onset of cervical cancer but is also a risk factor for breast cancer [73], and infection with HPV can cause apoptosis. Fibroblasts clearing apoptotic breast cancer cells infected with human papillomaviruses can uptake genetic material from these [74] in a Ca^2+^-dependent manner [19] (Figure 2F). The horizontal gene transfer between papillomavirus-infected cells and fibroblasts leads to significant changes in the fibroblasts, including increased colony formation ability and proliferation, resulting in malignant transformation of the fibroblasts [74]. While this nomenclature differs from the transformation of fibroblasts into CAFs, the properties are similar, and it is possible that these two cell types partially overlap. Additionally, a mechanism for horizontal gene transfer between cancer cells and fibroblasts is of note, as it promotes genetic instability in fibroblasts and may contribute to CAF formation.

### 3.4. Wound Healing

Fibroblasts in aberrant wound healing have been recognized as congruent with fibroblasts in tumors for decades [75]. In both scenarios, fibroblasts undergo activation, leading to increased proliferation; expression of shared markers, such as αSMA; and secretion of ECM components, including collagen and fibronectin [75,76]. Given this overlap, it is worth exploring the role of calcium signaling in wound healing and particularly fibrotic, aberrant wound healing.

Hypertrophic scar (HS) formation is a type of aberrant wound healing that is heavily dependent on fibroblast activation [77,78]. A recent study showed that the mechanosensitive calcium channel Piezo1 was overexpressed in HS tissue and was critical for the activation of fibroblasts and the formation of HS [78]. Inhibiting Piezo1 with either siRNA or the pharmacological inhibitor GsMTx4 significantly reduced proliferation, [Ca^2+^]_CYT_, αSMA, and collagen protein expression in vitro. Additionally, GsMTx4 reduced the size of HS, collagen, and αSMA expression in a mouse model, leading to the hypothesis that Piezo1 may be important in CAF activation in tumor models, particularly given the dramatic changes in matrix stiffness that accompany tumorigenesis, although this has not yet been explored.

## 4. Conclusions

Cancer cells and CAFs exist in a complex and reciprocal signaling environment where cues from one cell compartment have dramatic effects on the growth and survival of the niche as a whole. The studies highlighted in this review demonstrate diverse roles for Ca^2+^ signaling in promoting tumor growth, as well as in the development of CAF features. Diverse mechanisms have been revealed in CAF–cancer Ca^2+^ signaling; however, TRPA1-mediated VEGF secretion has emerged as a consistent contributor to disease progression. Given the availability of pharmacological inhibitors, this channel is a potential therapeutic target, particularly in the treatment of prostate cancer. We also presented evidence supporting the transition from monoculture to co-culture models that more closely replicate the in vivo setting. There remain many unexplored avenues for Ca^2+^ signaling in CAF–cancer interactions that will continue to be revealed to the extent that CAFs remain at the forefront of TME research.

## Figures and Tables

**Figure 1 ijms-22-11366-f001:**
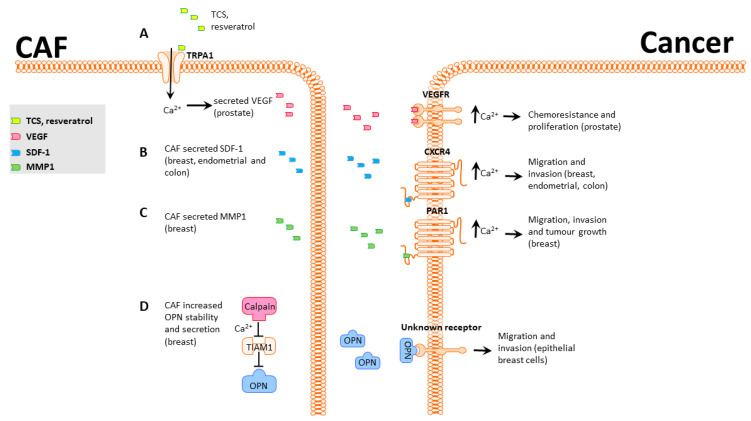
CAF-induced Ca^2+^-mediated phenotypic changes in cancer cells. (**A**), Prostate CAF TRPA1 can be activated by TCS or resveratrol, increasing secretion of VEGF. VEGFR activation in prostate cancer cells induces an increase in cytoplasmic calcium that promotes chemoresistance. (**B**), Breast, endometrial, and colon CAFs secrete SDF-1, which activates CXCR4 in cancer cells and induces an increase in [Ca^2+^]_CYT_ that promotes migration and invasion. (**C**), Breast CAFs secrete MMP1 and cleave and activate PAR1, leading to increased [Ca^2+^]_CYT_ and enhanced migration, invasion, and tumor growth in breast cancer cells. (**D**), Breast CAFs degrade TIAM1 in a Ca^2+^-dependent manner. This increases levels of the secreted factor OPN1, which activates an unknown receptor in breast cells and promotes migration and invasion.

**Figure 2 ijms-22-11366-f002:**
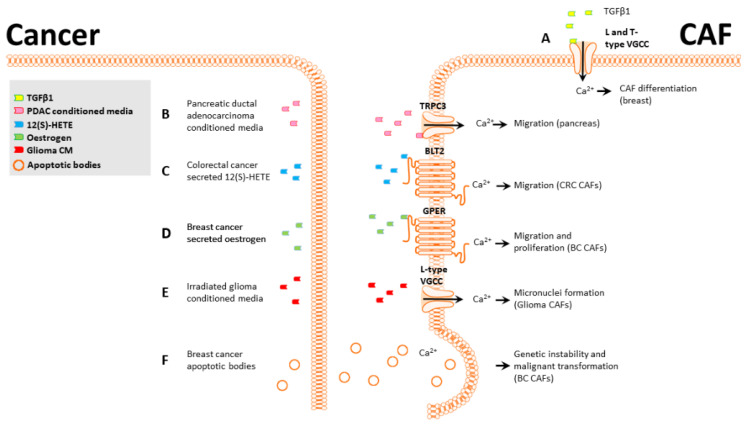
Ca^2+^-mediated phenotypic changes in CAFs. (**A**), TGFβ1-induced breast CAF differentiation is partially mediated by L and T-type VGCCs. (**B**), PDAC-conditioned media activate CAF TRPC3, and the Ca^2+^ influx promotes CAF migration. (**C**), CRC-secreted 12(S)-HETE activates the CAF BLT2 receptor and induces an increase in [Ca^2+^]_CYT_. (**D**), Breast cancer-secreted estrogen activates CAF GPER, increasing [Ca^2+^]_CYT_ and promoting CAF migration and proliferation. (**E**), Irradiated glioma-conditioned media activate CAF L-type VGCCs, and the Ca^2+^ influx promotes genetic instability and micronuclei formation. (**F**), Breast CAFs can phagocytose breast cancer apoptotic bodies in a Ca^2+^-dependent manner; this promotes horizontal gene transfer and malignant transformation in CAFs.

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
