# Peer review of "An Emerging Role for Calcium Signaling in Cancer-Associated Fibroblasts"

_ijms, 2021, doi:10.3390/ijms222111366_

Round 1
Reviewer 1 Report
This short review is clear and takes stock of the role of calcium signaling in CAF.
However, probably due to the brevity of the text, it is understandable that the authors do not cite a lot of papers, but it is a pity that very often they cite very old papers. They should also cite recent papers. Thus in addition to ref 23, they could cite Suzuki et al (Nat Comm 2014, 13;5:4153. doi: 10.1038/ncomms5153) who describe a family of genetically encoded Ca2+ indicators (CEPIA), and which can be used for the intra-organellar dynamics of Ca2+. Other references (20, 22, 24, 26, 28, etc.) could be supplemented by more recent references (very easy to find on PubMed ...).
The authors could clarify that calcium is pumped into the ER by SERCA and extruded by PMCA (P2 1.3. Ca2+ signal, end of 1st paragraph). Likewise, specify that the calcium signal is not only organize in time but is also organize in space (P2. beginning of the 3rd paragraph).
The authors rightly point out that the results obtained on single cell cultures can be very different from those obtained in vivo. In addition to the co-cultures that they mention in their review, they should also mention the use of organoids or animal models, especially these two important papers : Cancer Res. 2019 79(12):3139-3151 and Leung et alJ Clin Invest. 2018;128(2):589-606 respectively.
Finally, the figures, although clear, must be enlarged to be more readable.
Author Response
We thank Reviewer #1 for their comments and positive feedback. We have addressed their comments point by point below. Based on the reviews received, we have incorporated more and newer references in our manuscript as well as clarified and expanded sections on calcium signaling, CAF models and improved the visibility of the figures. Changes in the manuscript appear as track changes.
Reviewer 1
This short review is clear and takes stock of the role of calcium signaling in CAF.
However, probably due to the brevity of the text, it is understandable that the authors do not cite a lot of papers, but it is a pity that very often they cite very old papers. They should also cite recent papers. Thus in addition to ref 23, they could cite Suzuki et al (Nat Comm 2014, 13;5:4153. doi: 10.1038/ncomms5153) who describe a family of genetically encoded Ca2+ indicators (CEPIA), and which can be used for the intra-organellar dynamics of Ca2+. Other references (20, 22, 24, 26, 28, etc.) could be supplemented by more recent references (very easy to find on PubMed ...).
Thank you for the comments and suggestions for additional references. We have added the references specified (Suzuki et al, reference 26) on page 2 line 68. We have also supplemented our references that only cited older papers with more recent references for the references suggested as well as others.
Page 1, line 36 – added new reference (reference 10) (Baghban, Roshangar et al. 2020)
Page 2, line 63 – added new reference (reference 23) (Parker, Evans et al. 2017)
Page 2, line 78 – added new reference (reference 29) (Dolphin 2018)
Page 2, lines 80 and 91 – added new reference (reference 32) (Santoni, Morelli et al. 2020)
Page 2, line 91 – added new reference (reference 35) (Patergnani, Danese et al. 2020)
Page 3, line 122 – added new reference (reference 38) (Li, Wang et al. 2019)
The authors could clarify that calcium is pumped into the ER by SERCA and extruded by PMCA (P2 1.3. Ca2+ signal, end of 1st paragraph). Likewise, specify that the calcium signal is not only organize in time but is also organize in space (P2. beginning of the 3rd paragraph).
We have clarified SERCA (page 2, line 84) and PMCA (page 2 line 71) as well as the importance of spatial information in Ca2+ signalling (page 2, line 87).
The authors rightly point out that the results obtained on single cell cultures can be very different from those obtained in vivo. In addition to the co-cultures that they mention in their review, they should also mention the use of organoids or animal models, especially these two important papers : Truong et al., Cancer Res. 2019 79(12):3139-3151 and Leung et al., J Clin Invest. 2018;128(2):589-606 respectively.
Thank you for the comment, we have added reference to the organoid model on page 4 line 132 and cited the Truong et al paper (reference 40).
“Ongoing work is improving simple co-culture models to more accurately reflect the complexity observed in vivo by using microfluidics and patient-derived samples [40].”
We already describe animal models in this section (page 3, line 122) However, we agree that Leung et all is an important paper and now cite this on page 4, line 134 (reference 41).
“as well as in vivo models examining the roles of CAFs on chemoresistance through interactions with other components of the TME [41].”
Finally, the figures, although clear, must be enlarged to be more readable.
We have enlarged the figures. The original files are also available for your, and readers’, access to zoom in on specific sections as required.
References
Baghban, R., L. Roshangar, R. Jahanban-Esfahlan, K. Seidi, A. Ebrahimi-Kalan, M. Jaymand, S. Kolahian, T. Javaheri and P. Zare (2020). "Tumor microenvironment complexity and therapeutic implications at a glance." Cell Communication and Signaling 18(1): 59.
Dolphin, A. C. (2018). "Voltage-gated calcium channels: Their discovery, function and importance as drug targets." Brain and Neuroscience Advances 2: 2398212818794805.
Li, D., G. Wang, G. Jin, K. Yao, Z. Zhao, L. Bie, Y. Guo, N. Li, W. Deng, X. Chen, B. Chen, Y. Liu, S. Luo and Z. Guo (2019). "Resveratrol suppresses colon cancer growth by targeting the AKT/STAT3 signaling pathway." Int J Mol Med 43(1): 630-640.
Parker, I., K. T. Evans, K. Ellefsen, D. A. Lawson and I. F. Smith (2017). "Lattice light sheet imaging of membrane nanotubes between human breast cancer cells in culture and in brain metastases." Scientific Reports 7(1): 11029.
Patergnani, S., A. Danese, E. Bouhamida, G. Aguiari, M. Previati, P. Pinton and C. Giorgi (2020). "Various Aspects of Calcium Signaling in the Regulation of Apoptosis, Autophagy, Cell Proliferation, and Cancer." International journal of molecular sciences 21(21): 8323.
Santoni, G., M. B. Morelli, O. Marinelli, M. Nabissi, M. Santoni and C. Amantini (2020). "Calcium Signaling and the Regulation of Chemosensitivity in Cancer Cells: Role of the Transient Receptor Potential Channels." Adv Exp Med Biol 1131: 505-517.
Reviewer 2 Report
The review "An emerging role for calcium signalling in cancer-associated fibroblasts" written by Francisco Sadras and colleagues brings current and new knowledge about the role of Ca2+ signalling in the communication between cancer cells and associated fibroblasts.
This review is well written. However I would suggest going more deeper in the downstream molecular pathways that are modulated by this Ca2+ signaling?
For example, is it not clear what is the role of the ER in this signaling? Are there any crosstalk between ER-mito or other organelles when membrane Ca2+ channels are activated? This could be discussed in a new paragraph and also be outlined in the figure.
The section with CAFs genetic instability is not very well explained? What is genetic instability? What are the consequences in terms of cancer progression etc...?
Author Response
We thank Reviewer #2 for their comments and positive feedback. We have addressed their comments point by point below. Based on the reviews received, we have clarified and expanded sections on calcium signaling, genetic instability and downstream signaling. Changes in the manuscript appear as track changes.
Reviewer 2
The review "An emerging role for calcium signalling in cancer-associated fibroblasts" written by Francisco Sadras and colleagues brings current and new knowledge about the role of Ca2+ signalling in the communication between cancer cells and associated fibroblasts.
This review is well written. However I would suggest going more deeper in the downstream molecular pathways that are modulated by this Ca2+ signaling?
We have gone through and added more details on molecular pathways in the following sections (additions underlined). However, in many instances the scientists have not assessed downstream signaling as part of their published work, so we are unable to provide these details.
Page 4, line 153 “CXCR4 activation in cancer cells promotes Ca2+-dependent migration [45-47], possibly through PI3K/Akt and MAPK/Erk signaling.”
Page 6, line 246 “CAFs expressing GPER exposed to oestrogen or the GPER agonist G1 display increased intracellular Ca2+ as well as increased proliferation and migration through GPER/EGFR/ERK signaling.”
For example, is it not clear what is the role of the ER in this signaling? Are there any crosstalk between ER-mito or other organelles when membrane Ca2+ channels are activated? This could be discussed in a new paragraph and also be outlined in the figure.
We appreciate this comment and have expanded on the refilling of the ER during SOCE on page 2, line 84. While mitochondrial cross talk is an interesting area, it is a relatively new and complex field, and there has been very limited work done on this in CAFs with most publications focusing on cytosolic calcium changes. As none of the CAF publications in our review discuss or measure calcium in the mitochondria we believe extensive discussion on this might confuse the reader if we present concepts that are not further explored in the review. We also believe this complexity is beyond the scope of a non-calcium signaling focused review. However, to acknowledge the importance of this communication we now briefly discuss organelle cross talk on page 2 line 72.
“As well as calcium entry into the cytoplasm, intra-organelle communication is an emerging field and is one area that should be the focus of future studies. To date, studies have primarily been conducted in cancer cells [27] while less has been done in CAFs.”
The section with CAFs genetic instability is not very well explained? What is genetic instability? What are the consequences in terms of cancer progression etc...?
We have expanded and clarified the section on genetic instability on page 6 line 257.
“A potential complication from irradiation therapy is genetic instability arising from the bystander effect. Genetic instability can contribute to metastasis and chemoresistance by increasing genetic diversity within a tumor population [1].”
References
- Turajlic, S., et al., Resolving genetic heterogeneity in cancer. Nature Reviews Genetics, 2019. 20(7): p. 404-416.
Round 2
Reviewer 2 Report
Accept in present form